# CO$_2$ fixation by anaerobic non-photosynthetic mixotrophy for improved carbon conversion

Shawn W. Jones[1],*, Alan G. Fast[2],*, Ellinor D. Carlson[2],*, Carrissa A. Wiedel[1], Jennifer Au[3], Maciek R. Antoniewicz[3], Eleftherios T. Papoutsakis[2] & Bryan P. Tracy[1]

Maximizing the conversion of biogenic carbon feedstocks into chemicals and fuels is essential for fermentation processes as feedstock costs and processing is commonly the greatest operating expense. Unfortunately, for most fermentations, over one-third of sugar carbon is lost to CO$_2$ due to the decarboxylation of pyruvate to acetyl-CoA and limitations in the reducing power of the bio-feedstock. Here we show that anaerobic, non-photosynthetic mixotrophy, defined as the concurrent utilization of organic (for example, sugars) and inorganic (for example, CO$_2$) substrates in a single organism, can overcome these constraints to increase product yields and reduce overall CO$_2$ emissions. As a proof-of-concept, *Clostridium ljungdahlii* was engineered to produce acetone and achieved a mass yield 138% of the previous theoretical maximum using a high cell density continuous fermentation process. In addition, when enough reductant (that is, H$_2$) is provided, the fermentation emits no CO$_2$. Finally, we show that mixotrophy is a general trait among acetogens.

[1] White Dog Labs, Inc., 15 Reads Way, New Castle, Delaware 19720, USA. [2] Department of Chemical and Biomolecular Engineering and the Delaware Biotechnology Institute, University of Delaware, 15 Innovation Way, Newark, Delaware 19711, USA. [3] Department of Chemical and Biomolecular Engineering and Metabolic Engineering and Systems Biology Laboratory, University of Delaware, 150 Academy Street, Newark, Delaware 19716, USA. * These authors contributed equally to this work. Correspondence and requests for materials should be addressed to B.P.T. (email: btracy@whitedoglabs.com).

The production costs for most chemicals via microbial fermentation are currently high compared to oil-derived products primarily because of operating costs associated with feedstock and feedstock processing. Consequently, first and second generation bioproduct manufacturing processes are economically challenged, particularly in light of recent low oil prices. One way to mitigate high feedstock cost is to maximize conversion into the bioproduct of interest. This maximization, though, is limited because of the production of $CO_2$ during the conversion of sugar into acetyl-CoA in traditional fermentation processes. Acetyl-CoA is a central building block and a link between glycolysis and almost all downstream metabolic pathways and serves as a focal point for the production of biofuels and industrial chemicals by microbial fermentations. However, the ability to achieve metabolically efficient production of acetyl-CoA is hindered by energetic requirements and biochemical pathway constraints, requiring the production of $CO_2$ for every acetyl-CoA produced from glycolysis. Thus, one-third of all carbon in the feedstock is lost to $CO_2$, resulting in maximum carbon conversion of 67% at best, and lower in actuality due to cell mass creation, cell maintenance needs and other constraints[1].

It was previously demonstrated that a synthetic, non-oxidative glycolysis (NOG) pathway[2,3] enables the stoichiometric conversion of certain sugars to acetyl-CoA; however, NOG does not generate adenosine triphosphate (ATP) without converting acetyl-CoA into acetate, and consumes all reducing equivalents (that is, NAD(P)H) produced from sugar. Consequently, NOG-based product yields are limited by both ATP and NAD(P)H when producing metabolites that are more reduced, on a carbon basis (that is, carbon degree of reduction), than the feedstock consumed.

A second, alternative approach that stoichiometrically converts sugar to acetyl-CoA is anaerobic, non-photosynthetic mixotrophic fermentation[4] (here referred to as mixotrophy). Mixotrophy is defined as the concurrent utilization of organic (for example, sugars) and inorganic (for example, $CO_2$, CO, and $H_2$) substrates for growth and metabolism. As reviewed[4,5], the Wood–Ljungdahl Pathway (WLP), the carbon fixation pathway employed by acetogens to convert $CO_2$:$H_2$, CO or other C1 feedstocks into acetyl-CoA[6], is particularly well-suited for mixotrophy because it exhibits a low ATP requirement relative to other carbon fixation pathways[5] and requires the exact amount of NAD(P)H generated through glycolysis to fix two molecules of $CO_2$ into one acetyl-CoA[7]. Thus, one mole of hexose sugar yields three moles of acetyl-CoA and one mole of ATP through WLP-driven mixotrophy. With their ability to utilize gases through the WLP and a broad array of other carbohydrate substrates[7], acetogens are an ideal host organism for mixotrophy.

The amount of $CO_2$ re-assimilated with mixotrophy depends upon the degree of reduction of the desired metabolite (product). The more reduced the product, the less $CO_2$ can be re-assimilated, because NAD(P)H is directed towards product formation rather than $CO_2$ fixation. However, this reducing equivalent deficiency can be overcome through $H_2$-enhanced mixotrophy, whereby sufficient $H_2$ is exogenously provided to fully recapture the $CO_2$ lost in glycolysis. To avoid $CO_2$ emissions associated with $H_2$-production, electrolysis of water powered by solar, wind or hydroelectricity would be a preferred source and has achieved a level of maturity and success[8,9]. Alternatively, syngas can be added to sugar fermentation to provide the necessary reducing power and carbon. As reviewed[4,5,10], WLP gas-only fermentation is an ATP-limited process commonly requiring production of acetate from acetyl-CoA, for which acetogens receive their name, to generate sufficient ATP for cell growth and maintenance. Syngas-enhanced mixotrophy mitigates this challenge by supplying cells with abundant ATP through glycolysis.

In this study, we demonstrate the ability of a broad range of acetogenic organisms to conduct mixotrophy and $H_2$- or syngas-enhanced mixotrophy without carbon catabolite repression (CCR), thus enabling sugar to metabolite yields that are not theoretically possible through heterotrophic (that is, traditional) fermentation. Additionally, we demonstrate the ability to produce reduced products without the need for significant co-production of acetate, as is commonly witnessed in autotrophic fermentations[10,11]. Moreover, we show that sugar can be stoichiometrically converted to reduced products with nearly no $CO_2$ production from glycolysis. Last, we demonstrate the utility of mixotrophy in acetone production using a genetically-engineered acetogen (Clostridium ljungdahlii, abbreviated as CLJ). Acetone, a commodity petrochemical currently produced through the cumene process, has a world market on the order of six million metric tons per year which is valued at nearly eight billion US dollars. This makes acetone an attractive target for renewable, biochemical production given market drivers for bioderived acetone. When grown in a high cell density continuous fermentation system under mixotrophic conditions, we could achieve 138% of the theoretical, heterotrophic fermentation maximum, which is 92% of the theoretical mixotrophic maximum, and this was accomplished with volumetric productivities over $2 \, g \, l^{-1}$ per hour with acetone titers greater than $10 \, g \, l^{-1}$.

## Results

**Concurrent utilization of gas and sugar**. A key concern regarding the implementation of mixotrophy (Fig. 1a) is the possibility of CCR of the WLP in the presence of a preferred sugar substrate, as has been shown for two acetogens: Clostridium aceticum[12] and Blautia coccoides GA-1 (ref. 13). In contrast, Eubacterium limosum (ELM)[14] and Acetobacterium woodii[12] were found to concurrently utilize glucose and methanol and fructose and $CO_2$:$H_2$, respectively. To investigate if CCR occurs in CLJ, an important industrial strain[15,16], CLJ was grown on $C^{12}$-fructose and a $C^{13}$-labelled syngas mixture ($^{13}CO$:$^{13}CO_2$:$H_2$:$N_2$, 55:10:20:15) and interrogated at the metabolite, transcript and protein levels. CLJ incorporated a surprisingly large percentage of labelled gas into products (Fig. 1c). Between 73% and 80% of acetate, the primary metabolite, exhibited $^{13}C$-labelling over the course of the fermentation. Even at the earliest time point ($t = 24 \, h$), over 70% of acetate was derived from the labelled syngas rather than fructose. We also found no change at either the transcript or protein level of key WLP genes or enzymes, when compared against an autotrophic control (Supplementary Fig. 1), which is consistent with a previous report[17]. The closely related acetogen C. autoethanogenum (CAU) was also investigated under the same conditions, and we found it too displayed a high degree of $^{13}C$-labelled acetate, 51–58% labelled throughout the fermentation (Fig. 1d). Though it was not directly measured, ethanol, for both CLJ and CAU, should have a similar percentage of $^{13}C$-labelling, since it is also derived from acetyl-CoA. For both strains, the consistently high percentage of acetate labelling during fructose consumption demonstrates that both glycolysis and WLP operate concurrently, apparently without the limitation of CCR. To our knowledge, this is the first $^{13}C$-analysis of metabolites produced during mixotrophy.

**Engineered CLJ strain for acetone production**. We next sought to demonstrate the potential for mixotrophy to produce an industrially important metabolite at superior yields. Acetone was chosen as a proof-of-principle metabolite because it requires no additional NAD(P)H downstream of acetyl-CoA (Fig. 2a), which

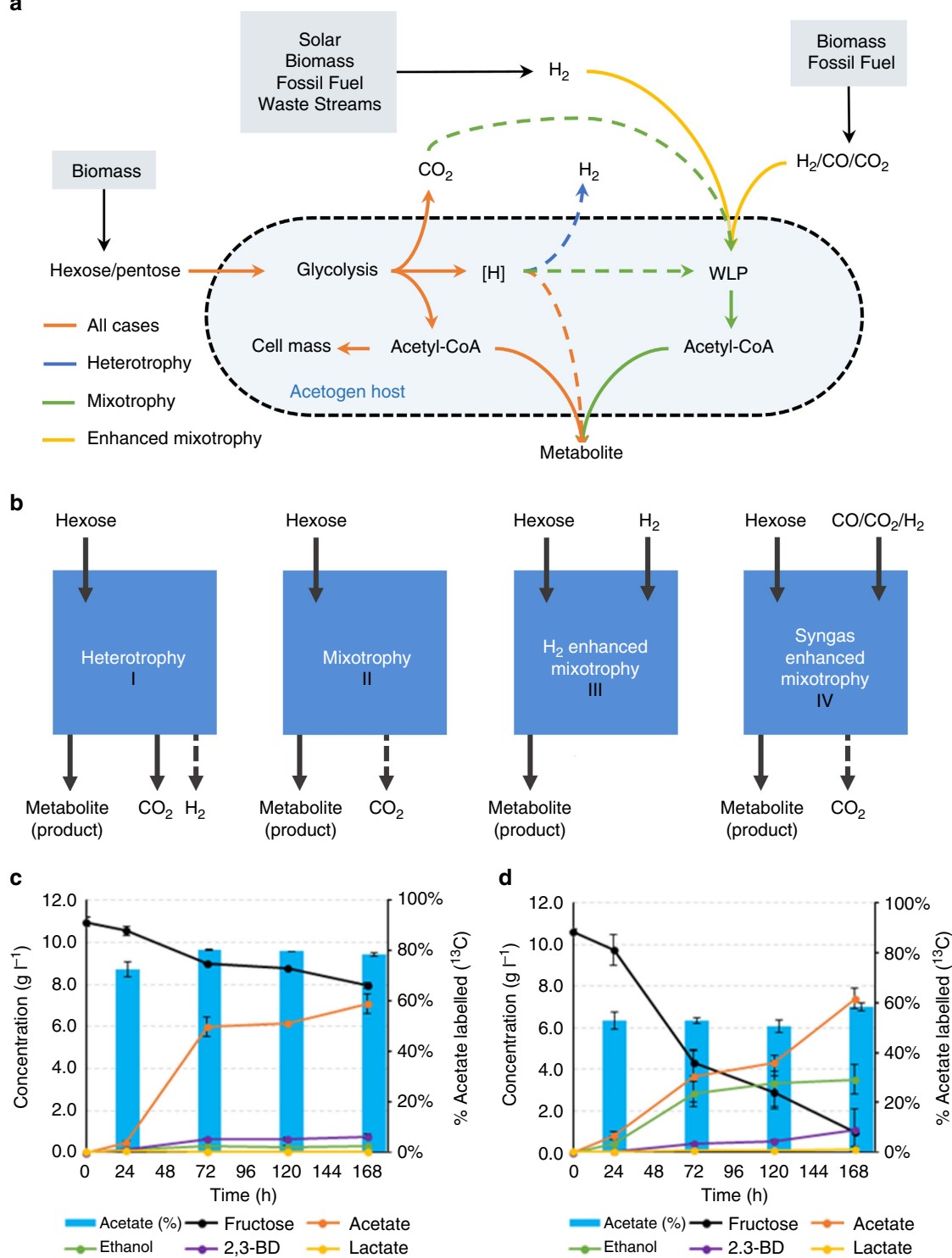

**Figure 1 | The concept of mixotrophy and its demonstration.** (**a**,**b**) Different modes of fermentation are shown as an abbreviated metabolic network (**a**) and block flow diagrams (**b**). Heterotrophy (case I): hexose is consumed and $CO_2$ and potentially $H_2$ are produced. Mixotrophy (case II): hexose is consumed and excess reducing equivalents are used to fix endogenously produced $CO_2$; any unconsumed $CO_2$ is released from the process. $H_2$-enhanced mixotrophy (case III): hexose along with $H_2$ are fed to the microorganism and no $CO_2$ is released. Syngas-enhanced mixotrophy (case IV): hexose and $CO:CO_2:H_2$ are fed to the microorganism. Depending on the composition of the syngas and the metabolite of interest, $CO_2$ may still be released from the process. Dashed lines indicate potential pathways or products. (**c**,**d**) [13]C-labelling fermentation profiles of CLJ (**c**) and CAU (**d**) during syngas-enhanced mixotrophy. Fructose (black line) consumed and metabolites produced during fermentation in the presence of a syngas mixture ([13]CO, [13]CO$_2$, $H_2$ and $N_2$). The percentage of acetate labelled with [13]C is shown in light blue for each time point. The s.d. of two biological replicates is shown in black error bars.

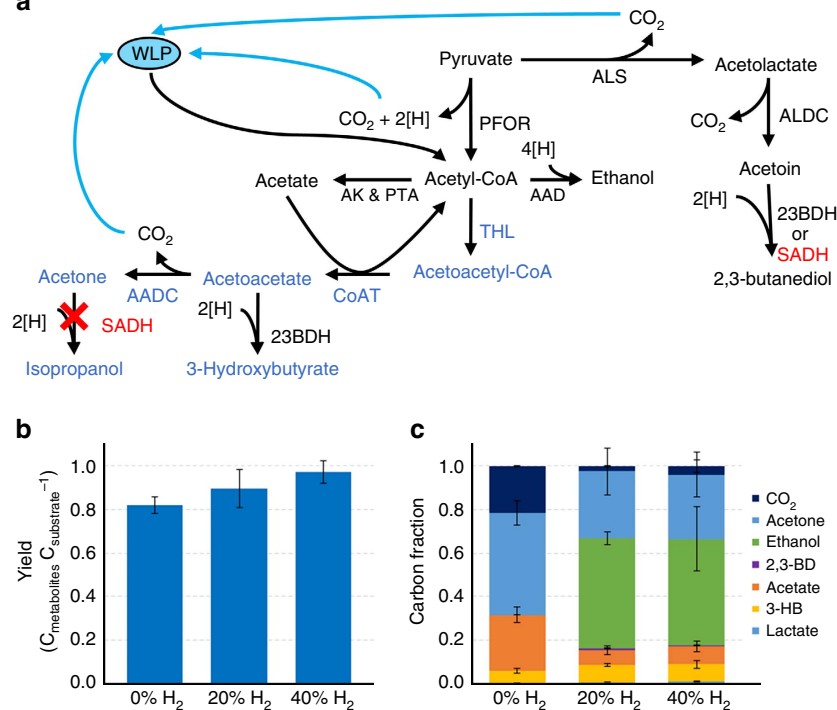

**Figure 2 | Metabolic engineering of CLJ to demonstrate mixotrophic production of acetone.** (**a**) Metabolic pathways downstream of pyruvate for the native and engineered CLJ. Native metabolites and enzymes are shown in black, and heterologous enzymes along with non-native metabolites are shown in blue. Integration with the WLP is shown through light blue arrows. The gene deletion is shown in red. PFOR, pyruvate:ferredoxin oxidoreductase; ALS, acetolactate synthase; ALDC, acetolactate decarboxylase; 23BDH, 2,3-butanediol dehydrogenase; SADH, secondary alcohol dehydrogenase; AK, acetate kinase; PTA, phosphotransacetylase; AAD, alcohol:aldehyde dehydrogenase; THL, thiolase; CoAT, CoA-transferase; AADC, acetoacetate decarboxylase. (**b,c**) Product profiles of CLJ ΔSADH (pTCtA) under mixotrophy and $H_2$-enhanced mixotrophy. Total molar yields (**b**) and product distributions (**c**) are shown. The s.d. of three biological replicates is shown in black error bars.

allows for all excess reducing equivalents to be used for $CO_2$ fixation. In addition, acetone is an important commodity chemical and a feedstock for poly(methyl methacrylate) production. Acetone is not a natural metabolite of any known acetogen, but recombinant production has been engineered into *C. aceticum*[18,19] and *A. woodii*[20]. However, both studies focused primarily on autotrophic growth where significant amounts of acetate were still produced, and although acetone titers and volumetric productivities demonstrate proof-of-principle, results lack industrial relevance. Additionally, an inducible acetone-producing strain of CLJ was previously constructed[21], though a recent publication[20] has questioned whether isopropanol was actually produced instead of acetone because of the native secondary alcohol dehydrogenase (SADH) activity of CLJ[22].

The SADH in CLJ is part of the 2,3-butanediol production pathway to convert acetoin into 2,3-butanediol and works in tandem with a 2,3-butanediol dehydrogenase (2,3-BDH; refs 22,23) (Fig. 2a). Importantly, this SADH was shown to convert exogenously added acetone into isopropanol[22]. To prevent this conversion, we deleted the SADH gene (CLJU_c24860) from the chromosome using a homologous recombination approach. We then constructed the acetone-producing plasmid pTCtA, consisting of a thiolase, a CoA-transferase, and an acetoacetate decarboxylase, and introduced it into the deletion strain CLJ ΔSADH. The resulting strain, CLJ ΔSADH (pTCtA), produced mostly acetone and acetate but also 3-hydroxybutyrate (3-HB) and trace amounts of isopropanol. We suspect 3-HB and isopropanol are still being produced to minor degrees because of the endogenous activity of 2,3-BDH, which is acting upon the intermediate acetoacetate to produce 3-HB or upon acetone to produce isopropanol (Fig. 2a).

After 168 h of mixotrophic growth, the total molar yield of CLJ ΔSADH (pTCtA) was 82% (Fig. 2b) with acetone being the primary metabolite at $1.73\,g\,l^{-1}$, followed by acetate at $1.54\,g\,l^{-1}$ and 3-HB at $0.29\,g\,l^{-1}$ (Fig. 2c, Supplementary Table 1). The mass yield of acetone was 34wt% and the mass yield of total acetone pathway products (acetone + 3-HB) was 37wt%. These mass yields are greater than the theoretical maximum acetone yield of 32wt% under standard heterotropic conditions, demonstrating the ability to improve mass yields of products with mixotrophy. However, the total mass yield of acetone pathway products (37wt%) is only 80% of the theoretical maximum under mixotrophic conditions. We, therefore, investigated a high cell-density continuous fermentation process for CLJ ΔSADH (pTCtA) to obtain higher acetone yields.

**High cell density continuous fermentation.** A potent approach to further improve product (metabolite) yields, minimize biomass formation, reduce ATP demand and increase the availability of reduction energy (available electrons) is through continuous fermentation with cell recycle. Cell recycle can also increase cell density, which has the potential to increase volumetric productivity[24]. To test this, we setup a high cell density continuous fermentation apparatus that fully retained cells until reaching an optical density at 600 nm ($OD_{600}$) of 35–60 (Fig. 3a), which corresponds to ~10–18 g l$^{-1}$ cell dry weight. As shown in Fig. 3a, the targeted cell densities were reached ~100 h after cell retention was started, at which point a harvest was implemented to maintain a constant cell density. Also at this point, the maximum acetone pathway (that is, acetone, 3-HB, and isopropanol) titer was achieved of $12.7\,g\,l^{-1}$ with concurrent acetate titers of $2.5\,g\,l^{-1}$. A minor amount of ethanol

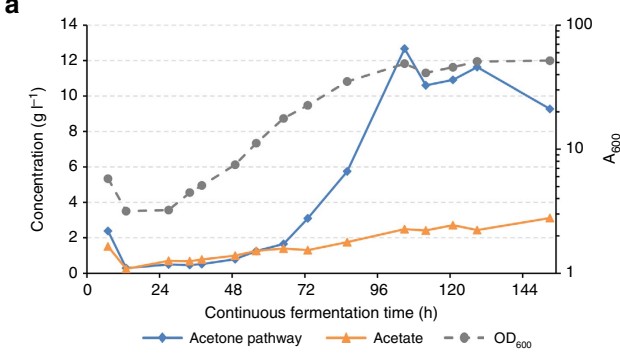

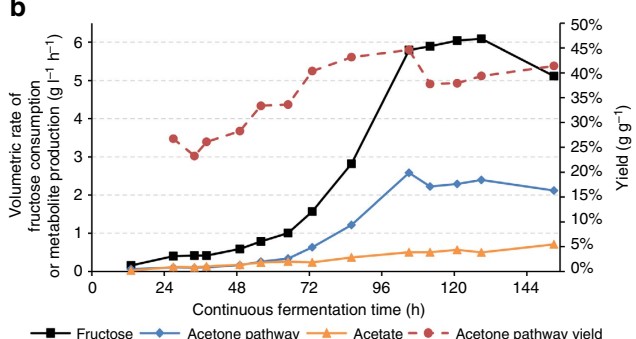

**Figure 3 | High cell density continuous acetone fermentation.**
Fermentation performance for 150 h of cell retention fermentation. At hour 96, a harvest was initiated to maintain a constant cell density. Titers and cell densities (**a**) are shown in addition to volumetric productivities and the total acetone pathway (acetone, 3-HB, and isopropanol) product yield (**b**). (**a**) Acetone pathway titers (blue diamond), acetate titers (orange triangle) and cell densities (grey circle with broken line). (**b**) Fructose volumetric consumption (black squares), acetone pathway volumetric productivity (blue diamonds), acetate volumetric productivity (orange triangles) and acetone pathway mass yield (%g g$^{-1}$) from consumed fructose (red circle with broken line). Individual product and fructose titers are shown in Supplementary Fig. 2.

(0.05–0.78 g l$^{-1}$, Supplementary Fig. 2) was also produced. Acetone concentrations reached a maximum of 10.8 g l$^{-1}$ (Supplementary Fig. 2), which was 85% of the total mass of acetone pathway products. Also shown in Fig. 3b, starting at hour 100 and for 50 h afterwards (that is, up to 150 h continuous fermentation time), volumetric productivities of acetone pathway products and acetate remained essentially constant, along with fructose volumetric consumption. The average volumetric productivity of acetone pathway products and acetate was 2.32 and 0.56 g l$^{-1}$ per hour, respectively, and the average fructose volumetric consumption rate was 5.78 g l$^{-1}$ per hour. This resulted in an average acetone pathway yield of 40.1wt% ($\sim$87% of the theoretical mixotrophic maximum). Real-time acetone pathway mass yields ranged from 37.8 to 44.6wt% during this time period (Fig. 3b). A biological replicate continuous fermentation performed similarly (Supplementary Fig. 3). Through subsequent process development, we have achieved 45wt% conversion of acetone pathway products with 5 g l$^{-1}$ per hour volumetric productivity and 23 g l$^{-1}$ titer of acetone pathway products with less than 3 g l$^{-1}$ acetate.

**H$_2$ enhanced mixotrophy leads to complete carbon utilization.** Though production of acetone does not require additional NAD(P)H, the pathway has a decarboxylation step (Fig. 2a), which generates one mole of CO$_2$ for every acetone mole produced. Under mixotrophic conditions, there is insufficient NAD(P)H to re-assimilate this additional CO$_2$, resulting in net CO$_2$ production (Fig. 2c, 0% H$_2$). This limitation can be overcome with H$_2$-enhanced mixotrophy, where H$_2$ provides the necessary reducing power to fix all CO$_2$. In the presence of either 20% or 40% (v v$^{-1}$) H$_2$, the overall molar yields of metabolites for CLJ ΔSADH (pTCtA) were >90% (Fig. 2b), and essentially all CO$_2$ was fixed into metabolites (Fig. 2c). However, under these highly reduced conditions, H$_2$ not only led to complete carbon capture but also caused the strain to favour reduced products, like ethanol, over less reduced products, like acetone. This caused a shift in the metabolite profile, with the primary metabolite now ethanol at 2.42 g l$^{-1}$ or 1.96 g l$^{-1}$, for 20% and 40%, respectively, and acetone being reduced to 1.02 g l$^{-1}$ or 0.81 g l$^{-1}$, respectively (Supplementary Table 1). Nevertheless, these data demonstrate the power of this technology to enable complete carbon utilization. Directing the carbon to the desirable product will require further strain engineering to eliminate the formation of undesirable products and using only the necessary amount of H$_2$ needed to achieve the zero CO$_2$ loss goal.

**Mixotrophy is a general trait of acetogens.** With the successful demonstration of mixotrophy in CLJ and CAU, we wanted to examine if mixotrophy was a more general trait among acetogens. Therefore, we tested two additional acetogens, *Moorella thermoacetica* (MTA) and ELM, for mixotrophy. ELM was previously shown to consume both glucose and methanol[14], the latter being utilized through the methyl or 'Eastern' branch of the WLP[6]. Here we wanted to test for concurrent sugar and gas utilization. All four strains (CLJ, CAU, MTA and ELM) were tested under both mixotrophic conditions and syngas-enhanced mixotrophic conditions. In addition, a heterotrophic control was prepared with the anaerobic bacterium *C. acetobutylicum* (CAC), which is related to many acetogens but has no native carbon fixation pathway. Under mixotrophy, all acetogens achieved carbon molar yields greater than the 67% theoretical maximum yield (C$_{\text{product moles}}$ C$_{\text{hexose consumed moles}}^{-1}$) that can be achieved without carbon fixation (Fig. 4a). The carbon yields for CAU and ELM were less than those for CLJ and MTA, presumably because CAU and ELM produced a larger percentage of reduced products (ethanol for CAU and butyrate for ELM, Fig. 4b), thus reducing the amount of CO$_2$ that can be fixed. Under syngas-enhanced mixotrophic conditions, the apparent carbon molar yields from sugar, as shown in Fig. 4a, for all acetogen species increased beyond the mixotrophic yields. In addition, the syngas mixture affected the product profiles with all strains, with more reduced products now being favoured (Fig. 4c). This shift occurs because of the added reducing power of the syngas mixture (55% CO and 20% H$_2$), which the cells are able to utilize to produce more reduced products. While acetate was still the primary metabolite, ethanol and 2,3-butanediol production increased in both CLJ and CAU, lactate production increased in MTA, and butyrate production increased in ELM (Fig. 4c). Demonstration of mixotrophy in these four strains, along with the demonstration in *A. woodii* previously[12], would argue that mixotrophy is a general trait of all acetogens, though to varying degrees.

**Mixotrophy benefits for other metabolites.** To highlight the potential of mixotrophic fermentations, we calculated the theoretical maximum yield for many metabolites of interest via mixotrophy and H$_2$-enhanced mixotrophy (Fig. 5a). Yield improvements are modest for more reduced products, such as ethanol and n-butanol, while yield improvements are significant for less reduced products including carboxylic acids. Generally, the ratio of mixotrophic to heterotrophic product yield is inversely proportional to the ratio of NAD(P)H to acetyl-CoA required

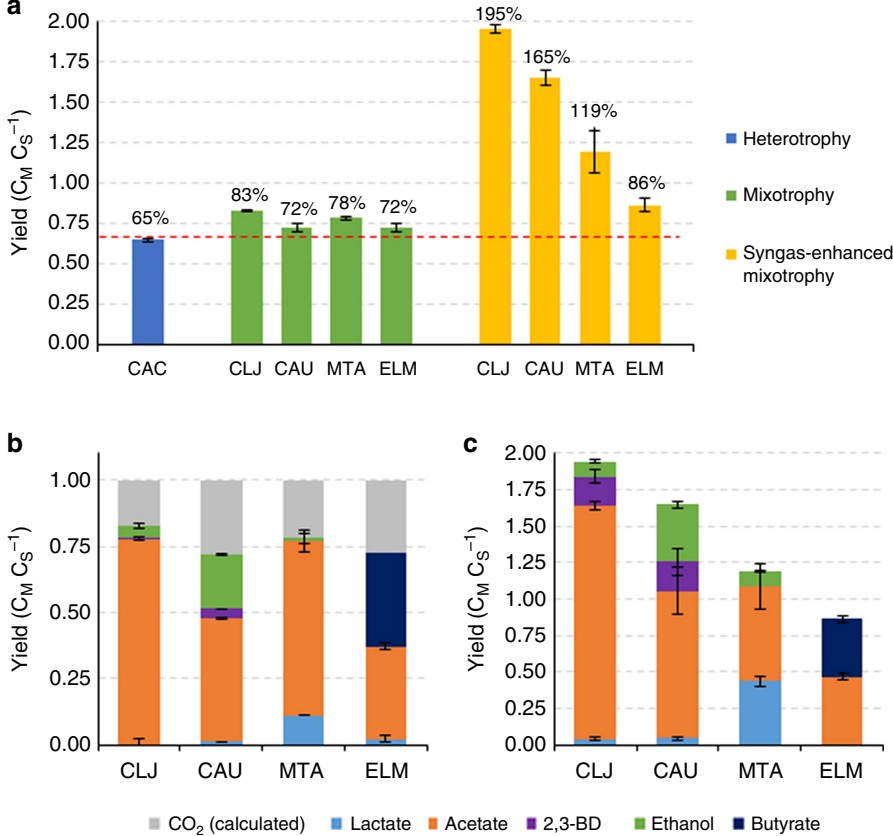

**Figure 4 | Mixotrophy is a general trait of acetogens.** (**a**) Carbon molar yields ($C_M \, C_S^{-1}$; $C_{metabolites} \, C_{substrate}^{-1}$), defined as the carbon moles produced divided by sugar carbon moles consumed, of *C. acetobutylicum* (CAC), *C. ljungdahlii* (CLJ), *C. autoethanogenum* (CAU) and *M. thermoacetica* (MTA) and *E. limosum* (ELM) grown under different conditions. Cultures were grown on fructose with either a $N_2$ headspace (heterotrophy and mixotrophy) or a syngas headspace (syngas-enhanced mixotrophy). The red dashed line indicates a yield of 67%. (**b,c**) Product profiles of CLJ, CAU, MTA and ELM under mixotrophy (**b**) or syngas-enhanced mixotrophy (**c**) are shown. Yields of $CO_2$ in **b** are calculated based on total carbon consumed for each strain. The s.d. of three biological replicates is shown in black error bars.

to produce a given chemical (Supplementary Table 2). Accordingly, the NAD(P)H to acetyl-CoA ratio can be used to quickly determine the potential yield improvement from mixotrophy for most metabolites with acetyl-CoA as a precursor (Table 1). With $H_2$-enhanced mixotrophy (Fig. 5a), product yields based on sugar mass increased 49–100% for all metabolites, aside from acetate, and the quantities of $H_2$ required for supplementation range from 0.017–0.065 g $H_2$ g glucose$^{-1}$ (Fig. 5b). As with the mixotrophic results, the chemicals that require the most reducing energy also require the largest amount of $H_2$ to fix all $CO_2$, and the amount of exogenous $H_2$ required is directly proportional to the NAD(P)H to acetyl-CoA ratio (Table 1). Since $H_2$-enhanced mixotrophy removes electron constraints, the chemicals that showed the smallest yield improvements from heterotrophy to mixotrophy show the largest yield improvements from mixotrophy to $H_2$-enhanced mixotrophy. For syngas-enhanced mixotrophy, similar yield improvements can be achieved using CO as the electron donor though $CO_2$ is generated in the process. In order to minimize net $CO_2$ production, a mixture of CO and $H_2$ is needed, and the ideal mixture is dependent upon the target metabolite. In this case, yields can be increased beyond those calculated here because the gaseous carbon can be fixed into metabolites, as seen in Fig. 4a,c.

## Discussion

Mixotrophic behaviour has been previously reported, such as the stoichiometric conversion of sugar into acetate by MTA[25] and the

co-consumption of sugar and C1 feedstocks for *A. woodii*[12] and *E. limosum*[14]. However, the potential of mixotrophy to increase product yields beyond previous theoretical limits, and its application to industrial biotechnology has largely been overlooked. Here we more thoroughly demonstrate that mixotrophy takes place and is a general trait of all four acetogens tested. Moreover, we genetically engineered a CLJ strain to produce acetone at a product yield that is 138% of the previous theoretical maximum, which was calculated based on heterotrophy alone. This significant increase in product yield could convert a previously uneconomical fermentation process into an economically viable one.

Though mixotrophy is a general trait among acetogens, allowing for the application of different strains for different metabolites or processes, the work in this study largely used CLJ as a mixotrophic host for several reasons. First, CLJ has proven to be genetically malleable, with several key genetic tools now available, including plasmid overexpression[15,16], inducible plasmid expression[21], chromosomal deletions[16,26] and chromosomal integrations[27]. The ability to genetically manipulate CLJ allows for the possibility to increase production of minor metabolites or introduce pathways to produce non-native metabolites, like acetone in this study. Another trait that makes CLJ a desirable mixotrophic host is the surprising result from the $^{13}$C-labelling analysis that in the presence of both sugar and gases, CLJ produced more acetate from the gaseous substrate than the sugar substrate. Indeed, over 70% of the produced acetate was

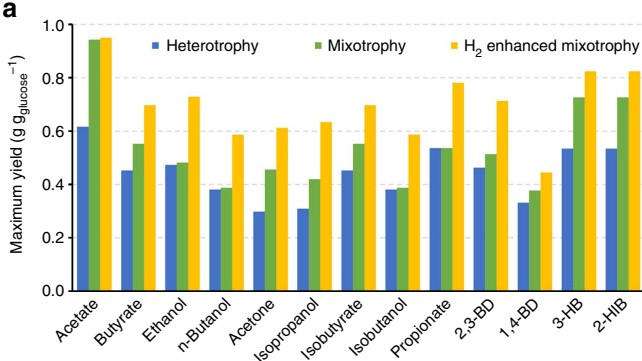

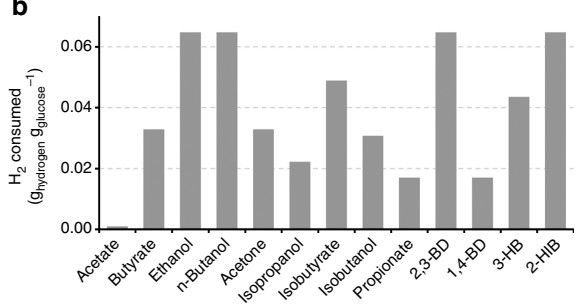

**Figure 5 | Theoretical yields for various metabolites to demonstrate the potential of mixotrophy.** (**a**) Calculated maximum biological mass yields for several metabolites of interest under heterotrophic, mixotrophic and $H_2$-enhanced mixotrophic conditions (where only enough $H_2$ is supplemented to consume all $CO_2$ evolved). Amounts of $H_2$ required to achieve these yields are shown in **b**. 2,3-BD, 2,3-butanediol; 1,4-BD, 1,4-butanediol; 3-HB, 3-hydroxybutyrate; 2-HIB, 2-hydroxyisobutyrate.

**Table 1 | Correlation between NAD(P)H:acetyl-CoA ratio to improvement in product yields with mixotrophy.**

| NAD(P)H: acetyl-CoA ratio | Increase in yield of mixotrophy over heterotrophy |
|---|---|
| 0 | 53% |
| 0.5 | 36% |
| 1.0 | 22% |
| 1.5 | 11% |
| 2 | 2% |
| 3 | 0% |

labelled with $^{13}C$ (Fig. 1c). Over the first 72 h, CLJ produced almost $6 g l^{-1}$ of acetate from about $2 g l^{-1}$ of consumed fructose. This is compared with the related CAU strain, which over the same period consumed $6.3 g l^{-1}$ of fructose and produced $3.6 g l^{-1}$ of acetate and $2.8 g l^{-1}$ of ethanol for a total of $6.4 g l^{-1}$ of product. In addition, the overall rate of fructose consumption in CLJ was significantly less than that in CAU. Over the 168 h fermentation, CLJ consumed only $3 g l^{-1}$ of fructose while CAU consumed over $9.6 g l^{-1}$ of fructose. One likely explanation for this divergent behaviour is the potentially inhibitory levels of acetate, which is a well-known inhibitor of many fermentations[28]. As shown in Fig. 1c,d, by 72 h, CLJ had produced almost $6 g l^{-1}$ of acetate while this titer was not reached in CAU until after 120 h. This will be further investigated to understand the differing behaviour of these two related strains.

Another important area that needs additional research and understanding is the effects the two types of substrates, organic and inorganic, have on enhanced mixotrophy. Here we tested only fructose with $H_2$ and fructose with a syngas mixture ($CO:H_2:CO_2:N_2$, 55:20:10:15), however many different organic and inorganic substrates can be used such as other sugars, gas compositions, methanol, formate, and other C1 substrates. Gaining a better understanding of the effect of all these possible substrates and their impact on product formation is crucial to best pair potential feedstocks for producing products of interest. An example of this effect can be observed in the syngas-enhanced mixotrophy results (Fig. 4). In those experiments, CLJ and CAU outperformed MTA and ELM in terms of total product yield from sugar (195 and 165% yields versus 119 and 86% yields, Fig. 4a). However, it is unclear whether these differences in performance are related to the strains themselves or rather the conditions tested. For example, the syngas composition may not have been ideal for MTA or ELM.

Beyond a fundamental analysis, the data from the high cell density continuous fermentation (Fig. 3) constitute a foundation for further development to achieve industrially relevant productivities, unprecedented mass yields and increased product titers.

## Methods

**Strains and growth conditions.** *C. ljungdahlii* DSM-13528 (CLJ), *C. autoethanogenum* DSM-10061 (CAU), *E. limosum* DSM-20543 (ELM) and *M. thermoacetica* DSM-521 (MTA) were obtained from DSMZ (Braunschweig, Germany). All cultures were grown in sealed serum bottles in a shaking incubator (150 r.p.m.) at 37 °C, except for MTA which was incubated at 55 °C. They were cultivated anaerobically in American Type Culture Collection (ATCC) medium 1754 with $10 g l^{-1}$ of fructose, with the exception of ELM which was grown in a modified ATCC medium 1754 (supplemented with $10 g l^{-1}$ MES, pH 6.0) with $5 g l^{-1}$ of fructose. Growth was monitored by measuring the optical density at 600 nm ($OD_{600}$). A 5% inoculum of mid-exponential phase ($OD_{600}$ of 0.8–1.5) was used to inoculate 160-ml serum bottles (Wheaton), with 50 ml of culture media and 110 ml of gas headspace. For autotrophic and syngas enhanced mixotrophic cultures, the headspace was pressurized to 30 psig with syngas ($CO:CO_2:H_2:N_2$, 55:10:20:15), except for ELM which was pressurized to 20 psig. MTA was grown in a mixture without CO ($CO_2:H_2$, 80:20). Heterotrophic and mixotrophic cultures were pressurized to 20 psig with $N_2$. The pH of the cultures was monitored and kept between 5.0–6.5 by adding 4M $NH_4OH$.

**$^{13}C$ Metabolite labelling.** CLJ and CAU were grown as described above with $10 g l^{-1}$ of $^{12}C$-fructose. The syngas-enhanced mixotrophic and autotrophic cultures contained a mixture of syngas ($^{13}CO:^{13}CO_2:H_2:N_2$, 55:10:20:15) with all carbons labelled. Samples were taken at early, mid and late exponential phase and metabolites and biomass was analysed for $^{13}C$ content. Gas chromatography–mass spectrometry analysis of $^{13}C$-labelling was performed on an Agilent 7890A GC system equipped with a DB-5 MS capillary column connected to a Waters Quattro Micro Tandem Mass Spectrometer operating under ionization by electron impact at 70 eV (ref. 29).

**Construction of CLJ ΔSADH (pTCtA).** The SADH gene in CLJ (CLJU_c24860) was deleted from the genome, as this enzyme readily converts acetone into isopropanol[22]. A replicating plasmid was constructed and transformed into CLJ to completely delete the gene using a homologous recombination technique. Two regions of homology, each about 1,400 bp in length, were PCR amplified from CLJ genomic DNA using the primers listed in Supplementary Table 3. These two regions were then PCR assembled, with a NotI restriction enzyme site in-between the two regions, and ligated into a pCR8:GW:TOPO TA entry plasmid (ThermoFisher Scientific). The chloramphenicol and thiamphenicol resistance gene from pSOS95-Cm was then PCR amplified, digested with NotI and ligated into the entry plasmid in between the two regions of homology. The integration cassette was then transferred from the entry plasmid to a custom destination plasmid using the Gateway LR Clonase II Enzyme mix (ThermoFisher Scientific). The custom destination plasmid was a modified pSOS95-based plasmid[30], where the expression cassette (including the transcriptional promoter and terminator) were replaced with the Gateway destination cassette and the repL episome was replaced with the episome from pNW33N (*Bacillus* Genetic Stock Centre, Columbus, Ohio).

The deletion plasmid was transformed into CLJ WT, using a protocol similar to the previously published protocol[16]. CLJ WT cultures were grown in 100 ml of ATCC medium 1754 with $5 g l^{-1}$ until an $OD_{600}$ of between 0.2–0.3. Cells were then harvested by centrifugation anaerobically and washed twice in anaerobic SMP buffer (270 mM sucrose, 1 mM $MgCl_2$, 7 mM sodium phosphate, pH 6). Cells were finally resuspended in 150 µl of antifreezing buffer (90:10 SMP: dimethylsulphoxide) and divided into 25 µl aliquots. Aliquots were mixed with 1 µg of plasmid DNA and then transferred to a 1 mm cuvette. Cells were pulsed at 625 V with a resistance of 600 Ω and a capacitance of 25 µF using a Gene Pulser Xcell

system (Bio-Rad). Cells were immediately mixed with 1 ml of ATCC 1754 medium and transferred to a 15 ml conical tube with 4 ml of ATCC 1754 medium. Cultures were incubated anaerobically at 37 °C for 16–20 h, until the cultures became turbid. Cells were then harvested by centrifugation and resuspended in 1 ml of ATCC 1754 medium. Four-hundred fifty microlitres of resuspended cells were then mixed with 25 ml of reinforced clostridial medium (RCM) molten agar (0.8%) with 5 μg ml$^{-1}$ thiamphenicol and poured into a petri dish. After the agar plates solidified, they were turned upside down and incubated anaerobically at 37 °C until colonies developed (3–5 days). Once developed, individual colonies were selected and grown in ATCC medium 1754 with 5 g l$^{-1}$ fructose and 5 μg ml$^{-1}$ thiamphenicol up to an OD$_{600}$ of about 1.0. They were then serial diluted and plated onto RCM plates with 5 μg ml$^{-1}$ thiamphenicol. Plates with individual colonies were replica plated onto fresh RCM plates with 40 μg ml$^{-1}$ erythromycin. Colonies which grew on thiamphenicol but were sensitive to erythromycin were selected, and deletion of CLJU_c24860 was confirmed via PCR and by sequencing of the local locus.

Plasmid pTCtA was constructed from a pSOS95-derived plasmid[30]. On the plasmid backbone, the repL episome was replaced with the episome from pNW33N and the expression cassette, including the transcriptional promoter, was removed except for the rho-independent terminator using a SbfI:KasI digestion. A synthetic acetone production operon was constructed consisting of the CLJ pta-ackA promoter (P$_{pta}$), the thiolase gene from *C. kluyveri* DSM 555 (CKL_3698, *thl*), the CoA-transferase gene from CAC (CA_P0163&0164, *ctfAB*), and the acetoacetate decarboxylase gene from CAC (CA_P0165, *adc*). All regions were amplified from genomic DNA using the primers listed in Supplementary Table 3. The synthetic operon was PCR assembled and then ligated into the modified backbone using the SbfI:KasI digestion. The plasmid was finally transformed into the CLJ ΔSADH and selected for using erythromycin (40 μg ml$^{-1}$ for solid agar plates and 100 μg ml$^{-1}$ for liquid media).

**Batch fermentations of CLJ ΔSADH (pTCtA).** Cultures of CLJ ΔSADH (pTCtA) were carried out in anaerobic ATCC medium 1754 with 5 g l$^{-1}$ fructose and 100 μg ml$^{-1}$ erythromycin in sealed 160-ml serum bottles, in a similar manner as described above. The atmosphere was either 30 psig N$_2$ or 30 psig of an H$_2$:N$_2$ mixture.

**High density cell recycle fermentations.** A 10% inoculum of mid-exponential phase (OD$_{600}$ of 0.8–1.5) CLJ ΔSADH (pTCtA) culture was used to inoculate a 3 l bioreactor (Applikon) with a 1.9 l working volume of anaerobic ATCC medium 1754 with 5 g l$^{-1}$ fructose and 5 μg ml$^{-1}$ of clarithromycin. Cultures were sparged with N$_2$ for the first 24 h to maintain an anaerobic environment as the cultures entered exponential growth. Once the cultures reached an OD$_{600}$ of 1.5, cell-recycle was initiated. Fermentation pH was controlled from dropping below 5.0 using 4M NH$_4$OH. Fermenter feeding and permeate rates were balanced to maintain a constant working volume and the average rate was 3–6 ml min$^{-1}$. Cell retention was accomplished with a 0.15 ft$^2$, 0.1 μM Graver Technologies (Newark, DE, USA) sintered stainless steel microfiltration membrane. The feed rate to the membrane was 150 ml min$^{-1}$ and the recirculation rate was 6.5 l min$^{-1}$. The retentate rate was between 144–147 ml min$^{-1}$, which equaled the membrane feed rate minus the permeate rate. The culture was in complete cell retention until reaching an OD$_{600}$ of 35–60, at which time a harvest was started that was equal to the critical dilution rate that maintained a constant cell density in the fermenter. Fructose concentration in the feed was increased to as high as 80 g l$^{-1}$ during the course of building cell density, and the concentration of clarithromycin was maintained at 5 μg ml$^{-1}$.

**Detection of metabolites.** Culture supernatant samples were quantified using an Agilent HPLC instrument with a Bio-Rad Aminex HPX 87H column using a 5 mM H$_2$SO$_4$ mobile phase[31]. Gas samples were analysed using an Agilent GC instrument with a Supelco 60:80 Carboxen-1,000 column, used according to the manufacturer's recommendation.

**RNA isolation and quantitative reverse transcription (qRT–PCR).** RNA was extracted using the RNEasy Kit (Qiagen) with a modified protocol as follows: cells were thawed on ice and washed and resuspended in 220 μl RNase-free SET buffer + 20 μl proteinase K. Samples were sonicated for 10 min using 15 s pulse on:off at 40% amplitude in a Fisher Scientific Sonic Dismembrator. In all, 1 ml of Trizol was added to each sample, split in two and another 400 μl Trizol was added to have the final volume of ~1 ml. 200 μl of ice-cold chloroform were added to the sample and mixed for 15 s, then incubated at room temperature for 3 min. Samples were spun at >12,000g for 15 min at 4 °C. Upper aqueous phase was mixed with 500 μl of 70% ethanol and RNEasy protocol was followed per manual. Sample concentration was determined using a NanoDrop (Thermo Scientific). Concentrations ranged from ~50 ng μl$^{-1}$–600 ng μl$^{-1}$ depending on the sample.

Complementary DNA (cDNA) was synthesized using high capacity cDNA reverse transcription kit (Applied Biosystems). In all, 1 μl of cDNA was used in qRT–PCR experiments using the SYBR green kit (BioRad) following the manufacturer's protocol. Primers are listed in Supplementary Table 2. Each sample was tested with three technical replicates and three biological replicates. All technical replicates were averaged. The housekeeping gene was CLJ_c12520 (*gp*k, guanylate kinase)[23]. Fold changes were calculated using the autotrophic condition as the control case and either mixotrophy or syngas-enhanced mixotrophy as the experimental condition. All calculations were done using Graph Pad PRISM software with statistical analysis using two-way analysis of variance of the ΔCT between the three conditions (autotrophy, mixotrophy and syngas-enhanced mixotrophy).

**Western blots.** Cells were harvested by centrifugation at 4 °C for 10 min at 7,000g. The cell pellets were frozen at −82 °C. The cell pellets were then resuspended in 500 μl of 100 mM Tris-HCl (pH 7.0) (Buffer A). The samples were sonicated using a Fisher Scientific Sonic Dismembrator for 15 s pulse on:off at 50% amplitude for 10 min. The samples were spun down for 10 min at 13,000g. The supernatant was collected and the cell debris was resuspended in same amount of Buffer A. The samples were mixed in a 1:5 dilution with 5× loading buffer (0.2 M Tris-HCl (pH 6.8), 10% SDS, 1.43M 2-mercaptoethanol, 0.05% bromophenol blue, 50% glycerol) and boiled at 95 °C for 10 min. A volume of 30–50 μl of aliquots were loaded onto the Genscript ExpressPlus PAGE Gel, 4–20% gradient and run at 120 V for 2 h. The gels were transferred using the BioRad mini-trans blot system. The custom polyclonal antibodies against the CODH subunit were designed through ProteinTech and obtained from rabbit serum. The AP colourimetric kit (Anti-Rabbit Assay Kit #1706460, BioRad) was used to visualize the bands using a 1:10,000 dilution of the polyclonal antibodies in TBS and Tween 20, following the supplier's recommendations.

**Stoichiometric and energetic model.** The stoichiometric and energetic model was created for combined glycolysis and WLP that maximizes production of a specific metabolite, while assuming 5% of all carbon goes towards biomass formation, as previously detailed[4]. Briefly, the model considers ATP and acetyl-CoA production according to equations 1 and 4. Equation 1 describes CO$_2$ fixation via the WLP. Included in equation 1 is an ATP conservation coefficient (*n*), which accounts for ATP generated through membrane-bound ATPases, driven by cytochrome- or Rnf-created H$^+$ or Na$^+$ membrane gradients. For the calculations presented here, $n = 0.9$. Equation 4 represents ATP, acetyl-CoA, CO$_2$ and reducing equivalent production from glycolysis of hexose sugar to acetyl-CoA. Cell production was defined according to equation 2. Reducing equivalent production from hydrogen was modelled according to equation 3.

$$2CO_2 + 4H_2 + ATP \rightarrow AcetylCoA + 2H_2O + nATP \qquad (1)$$

$$1/2 - AcetylCoA + (M_{cell}/Y_{ATP})ATP + 0.3NAD(P)H$$
$$\rightarrow C - mole\ Biomass\ (CH_{2.08}O_{0.53}N_{0.24}) \qquad (2)$$

$$H_2 + NAD(P)^+ \leftrightarrow NAD(P)H + H^+ \qquad (3)$$

$$1Hexose \rightarrow 2AcetylCoA + 2CO_2 + 2ATP + 4NAD(P)H \qquad (4)$$

$M_{cell}$ is the molecular weight of cell biomass, and a value of 26 g mol$^{-1}$ is used[32]. $Y_{ATP}$ is the ATP yield coefficient, defined as g$_{cells}$ mol$_{ATP}^{-1}$, and a value of 6.5 g$_{biomass}$ mol$_{ATP}^{-1}$ was used, as determined in ref. 33. C-mole Biomass (CH$_{2.08}$O$_{0.53}$N$_{0.24}$) is the molecular formula for biomass as determined in ref. 32. NAD(P)$^+$/NAD(P)H stands for nicotinamide adenine dinucleotide or nicotinamide adenine dinucleotide phosphate. NAD(P)$^+$ is the oxidized form, and NAD(P)H is the reduced form. H$^+$ is a proton. Subsequent metabolic equations were added for each metabolite analysed.

**Data availability.** All relevant data are included with the manuscript (as figure source data or supplementary information files), and all data is available upon reasonable request from the corresponding author.

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

## Acknowledgements

This work was supported by the US National Science Foundation STTR Phase I program under Award IIP-1346424.

## Author contributions

S.W.J. performed many of the fermentation experiments, analysed the data and wrote the paper. A.G.F performed some of the fermentation experiments, did the modelling analysis and wrote the paper. E.D.C. performed some of the fermentation experiments, the qRT–PCR, western blots and wrote the paper. J.A. and M.R.A performed the $^{13}$C analysis. E.T.P analysed data and wrote the paper. B.P.T. designed the study, analysed the data and wrote the paper. S.W.J, A.G.F. and E.D.C. contributed equally to the study. All authors discussed the results and commented on the manuscript.

## Additional information

**Competing financial interests**: White Dog Labs is commercializing a mixotrophy-based fermentation process and has filed a patent application PCT:US2016:019760.

**Reprints and permission** information is available online at http://npg.nature.com/ reprintsandpermissions/

**How to cite this article**: Jones, S. W. et al. $CO_2$ fixation by anaerobic non-photosynthetic mixotrophy for improved carbon conversion. *Nat. Commun.* **7:**12800 doi: 10.1038/ncomms12800 (2016).

