## [Peer Review File · Nature Communications]

Reviewer #1 (Remarks to the Author)

In this manuscript, the authors describe 'Anaerobic, Non-Photosynthetic (ANP)' mixotrophic fermentation as a tool for both increasing product conversion yields and reducing overall CO₂ emissions from fermentation processes. As a proof-of concept, the authors engineered *Clostridium ljungdahlii* to produce acetone at a mass yield 138% of the previous theoretical and, when enough reducing power (e.g., H₂) is provided, to avoid emission of CO₂. Overall, the quality of this work is great, with clear explanation of hypotheses and approaches and outstanding results. I think this will be a seminal paper in the field. Frankly, we are also working on similar topic, but I have to say that this is a solid work. I only have the following minor points.

[Minor comments]

- 1) Authors can include more discussion on the future direction of the work, as this will be of great interest to the readers.
- 2) Operation condition for continuous, high-cell density fermentation should be described in more detail.
- 3) In line 99, please represent the result of continuous, high-cell density fermentation with additional supplemental Figure, which will help to the readers understand.
- 4) Figure 2b should be cited prior to Figure 2c.
- 5) In Figures 2 and 3, A,B,(and C) should be a small letters.

Reviewer #2 (Remarks to the Author)

Manuscript title: CO₂ fixation by anaerobic, non-photosynthetic mixotrophy for improved carbon conversion

Manuscript number: NCOMMS-16-05946-T

The manuscript with title "CO₂ fixation by anaerobic, non-photosynthetic mixotrophy for improved carbon conversion" and number NCOMMS-16-05946-T for the journal Nature Communications is a proof of concept of engineering *Clostridium ljungdahlii* (CLJ) to produce acetone using different types of mixotrophy (without gas addition, with addition of H₂ and with addition of syngas).

The author selected to mainly focus on *Clostridium ljungdahlii*, which is a very interesting microorganisms do to its capacity to metabolize sugars and/or CO₂/H₂ and synthesis gas (CO/H₂). The latter feature makes it very interesting for the industrial biotech, reducing the dependency on crude oil, while also contributing to reduce CO₂ from the atmosphere.

My major concern is the novelty of the manuscript for a journal like Nature Communications. The novelty is to highlight that the synthesis capabilities of *C. ljungdahlii* from CO and CO₂ are not limited to only butanol and ethanol, but that they can be engineered to produce other compounds like acetone using synthetic biology tools.

The authors proved their hypothesis that syngas enhanced mixotrophy would deliver higher incorporation of syngas into products while avoiding the formation of undesired products like acetate.

Nevertheless the authors fail to mention previous work using acetogen *C. aceticum* where the native pathway from *C. acetobutylicum* was cloned to produce acetone independently from acetate. The authors also fail to mention a relevant paper review about pathway engineering and synthetic biology tools using acetogens, which covers a whole section for the production of acetone by acetogens (B. Schiel-Bengelsdorf and P. Durre, 2012, Pathway engineering and synthetic biology using acetogens, *Synthetic Biology*, 586, 2191-2198). In that review, previous works engineering acetogens to produce acetone up to 8 mg/L are described which were not mentioned in this manuscript, thus, proof of principle for using recombinant acetogens growing on CO₂/H₂ gas mixtures and producing important bulk chemicals has already been provided, so the

novelty is using CLJ as a proof of principle instead of *C. acetivum*. Nevertheless, the authors do not give any final acetone yield in mg/L and very few quantitative comparisons are given with previous works in literature, which really undermines the manuscript. Also, more rigorous results are needed to completely backup some claims from the authors.

Also, some results were only repeated twice, which I believe is not enough.

Based on this, I would not recommend the manuscript for publication in Nature Communications.

My other specific comments to improve the paper would be as follows:

- 1) I would add a nomenclature section to enhance the reading experience.
- 2) Adding a sentence in the introduction describing the main characteristic of acetogens would be useful
- 3) Adding a sentence in the introduction mentioning why acetone is an industrially relevant product to synthesize using microorganisms would be appropriate.
- 4) In the introduction, it would be worth to mention if any acetogens naturally produce acetone ? Or if they have been engineered to produce acetone ?
- 5) I was a little confused in this: In line 71 it is mentioned that *C. Ljungdahlii* was selected to determine if it didn't suffer from catabolite repression of the Wood-Ljungdahl pathway (WLP) in the presence of the preferred sugar fructose (in comparison with *C. acetivum* and *Blautia cocoides* which do suffer catabolite repression). The authors seem to conclude CLJ doesn't suffer from catabolite repression, based on the comparison of either transcript and protein levels with an auxotroph control. Nevertheless the results of the auxotroph control are not shown in Fig S1a (transcription studies), it would be good to add them.
- 6) Also, relevant to comment 4, in Figure S1b, only one protein of the whole WLP was studied using western blot. Why was this enzyme selected and why were not the other enzymes studied too? How do the authors know there was no translational repression of the other 3 enzymes ?
- 7) Relevant to comment 4 and 5, if it was indeed proven that CLJ did not suffer from catabolite repression, adding one sentence would be adequate to compare with results of the repressed strains? (maybe add an hypothesis of why CLJ may be different ?)
- 8) In line 71, what was the concentration of fructose used in the labeling experiments?. In the materials and methods it says 5 g/l (lines 269 and 274), but in figure 1 c and d, it seems fructose was 10 g/l at time 0.
- 9) In Line 73 it is mentioned that the syngas composition is (CO:CO₂: H₂: N₂: 55:10:20:15) : Why was this specific component concentration of syngas selected ? Was it based on some previous optimization ?
- 10) In Line 75 it is mentioned: "CLJ incorporated a surprisingly large percentage of labeled gas into products". Any hypothesis why this marked preference for syngas instead of fructose? How does it compare in literature with other mixotrophic acetogens ?
- 11) In line 84 and Figure 1 c and d, Why when using *C. autoethanogenum* (CAU) all fructose is consumed but not with CLJ ? Probably this is related to the comment 9. It would be nice to further explain, as this seems to be an industrially relevant feature from CLJ.
- 12) In line 84 and figure 1c and d: How was growth yield different between the two microorganisms in terms of growth yield etc ?

- 13) In line 75, How does the labeled gas incorporation into acetate compares with other labeling works made for acetogens mixotrophes or mixotrophes in general ?
- 14) In line 75 and line 81 and Figure Did the labeling for ethanol was also measured ? How does it compare with acetate?
- 15) In line 84 and figure 1c and d : any hypothesis would CAU produce more ethanol since both CAU and CLJ have the enzyme AAD to produce ethanol from Acetyl -CoA ?
- 16) In line 93, it was found that after knocking out the secondary alcohol dehydrogenase (SADH) coding gene, the resulting strain produced acetone and acetate but also high quantities of 3-hydroxybutyrate. Why was the gene of the enzyme 2,3BDH, 2,3- butanediol dehydrogenase not knocked out too to increase acetone yield ?
- 17) Line 95: Was it possible to measure the total production in g/L of acetate, and 3-HB ? Only carbon fractions are presented in Fig B and C.
- 18) In line 96:" 3-HB is presumably produced from the native 2,3- butanediol dehydrogenase (2,3BDH) acting upon the intermediate acetoacetate", adding a reference would be adequate.
- 19) In Line 99, while using the fermentor, what was the final cell density and final yield in g/L of acetone and 3-HB?
- 20) Line 105. The results of 20% and 40% H₂ look statistically the same, why didn't the authors try using 10%, which would have favored the production of acetone?
- 21) In line 124, "the syngas mixture affected the product profiles with all strains producing a greater portion of reduced products (Supplemental Fig. S2)". It would be interesting to add one more sentence to explain why ? It is especially interesting to see in Figure S2 the shift from microorganism ELM to produce butyrate instead of ethanol
- 22) In line 140, it would be interesting to compare the maximum biological mass yields predicted in Figure 3b under different mixotrophic condition with results from literature ?
- 23) In line 288, where de genes of thiolase, acetate/acetoacetate CoA-transferase and acetoacetate decarboxylase codon optimized ?
- 24) In line 290, how was the transformation achieved?
- 25) The pathway from Figure 1a is rather over simplified, something like the pathway shown in Figure 1a of the paper from Kopke et al. 2010 (the author is cited in this manuscript) would make it easier for the reader to understand the paper. Specially when enzymes like ACS/CODH, ACS alpha, ACS Beta, CFeSP alpha, CFeSP beta are mentioned in the paper but the full name is not given
- 26) In the discussion or conclusions, it would be relevant to add a sentence mentioning the key challenges to be addressed in order to improve yields using mixotrophy ?

Reviewer #3 (Remarks to the Author)

Jones et al. show that anaerobic, non-photosynthetic mixotrophy can both increase product conversion yields and reduce overall CO₂ emissions from fermentation processes. As a proof-of-concept, they engineered *Clostridium ljungdahlii* to produce acetone at a high mass yield and

demonstrated that when enough reductant (e.g., H₂) is provided no CO₂ was produced. They also show that mixotrophy is a general trait among most acetogens, and present an experimentally-validated model to predict yield increases for a variety of bioproducts of industrial interest. Although it is a very nice piece of work with interesting concepts, the main concern of the manuscript is that most of the experiments regarding acetone production by *C. ljungdahlii* cannot, in their current state of description, be reproduced:

- 1) How was the deletion of *sadh* made by homologous recombination (replicative/non replicative plasmid, use of PCR product? How was the DNA introduced in *C. ljungdahlii*?)
- 2) How was the pTcTA plasmid constructed? Was the *thIA* promoter from *C. acetobutylicum* conserved in the pSOS95 or was it replaced by the *thIA3* promoter from *C. kluyveri* ?
- 3) How was the high cell density culture performed? What were the bleeding and dilution rates? What was the mass yield of acetone formation (currently only the mass yield of acetone+3HB is given)? Although the mass yield is an important parameter, titer and productivity are also key for an industrial process and should also be provided.

Other comments:

-Line 90 It is claimed that *sadh* (deleted in the acetone producing strain) is naturally responsible for 2, 3 butanediol production. Then line 94 it is said that "3-HB is presumably produced from the native 2,3-butanediol dehydrogenase acting upon the intermediate acetoacetate ». How can *SadH* reduced acetoacetate if the encoding gene has been deleted?

-Mixotrophic growth with high mass yield of product formation has already been described for *Eubacterium limosum* and the reference should be added (Loubière, P., GROS, E. V. E. L. Y. N. E., Paquet, V., & Lindley, N. D. (1992). Kinetics and physiological implications of the growth behaviour of *Eubacterium limosum* on glucose/methanol mixtures. *Microbiology*, 138, 979-985.)

Point by point response to reviewers:

Reviewer #1 (Remarks to the Author):

In this manuscript, the authors describe 'Anaerobic, Non-Photosynthetic (ANP)' mixotrophic fermentation as a tool for both increasing product conversion yields and reducing overall CO₂ emissions from fermentation processes. As a proof-of concept, the authors engineered *Clostridium ljungdahlii* to produce acetone at a mass yield 138% of the previous theoretical and, when enough reducing power (e.g., H₂) is provided, to avoid emission of CO₂. Overall, the quality of this work is great, with clear explanation of hypotheses and approaches and outstanding results. I think this will be a seminal paper in the field. Frankly, we are also working on similar topic, but I have to say that this is a solid work. I only have the following minor points.

[Minor comments]

1) Authors can include more discussion on the future direction of the work, as this will be of great interest to the readers.

Great suggestion. The previous manuscript version was a very short format and transferred without editing to Nature Communications. So details and discussion were severely limited. We have now included this in the discussion section on pages 12-14.

2) Operation condition for continuous, high-cell density fermentation should be described in more detail.

We agree and admit that this was a significant oversight on our behalf. We have added a section in the Methods titled "High Density Cell Recycle Fermentations." We have also expanded upon the results on pages 8 and 9 (lines 152-175) and added in the main text Figs. 3a&b and supplemental Figs S2&3.

3) In line 99, please represent the result of continuous, high-cell density fermentation with additional supplemental Figure, which will helpful to the readers understand.

Please see our response to comment #2 above.

4) Figure 2b should be cited prior to Figure 2c.

Thank you for pointing this out, and we have revised the manuscript accordingly.

5) In Figures 2 and 3, A,B,(and C) should be a small letters.

Thank you for pointing this out. We have revised the figures accordingly.

Reviewer #2 (Remarks to the Author):

Manuscript title: CO₂ fixation by anaerobic, non-photosynthetic mixotrophy for improved carbon conversion

Manuscript number: NCOMMS-16-05946-T

The manuscript with title "CO₂ fixation by anaerobic, non-photosynthetic mixotrophy for improved carbon conversion" and number NCOMMS-16-05946-T for the journal Nature Communications is a proof of concept of engineering *Clostridium ljungdahlii* (CLJ) to produce acetone using different types of mixotrophy (without gas addition, with addition of H₂ and with addition of syngas).

The author selected to mainly focus on *Clostridium ljungdahlii*, which is a very interesting microorganism due to its capacity to metabolize sugars and/or CO₂/H₂ and synthesis gas (CO/H₂). The latter feature makes it very interesting for the industrial biotech, reducing the dependency on crude oil, while also contributing to reduce CO₂ from the atmosphere.

My major concern is the novelty of the manuscript for a journal like Nature Communications. The novelty is to highlight that the synthesis capabilities of *C. ljungdahlii* from CO and CO₂ are not limited to only butanol and ethanol, but that they can be engineered to produce other compounds like acetone using synthetic biology tools.

We respectfully but strongly disagree with the assertion that the novelty of our manuscript is as the reviewer suggests. The novelty of this work is NOT about producing acetone (or any other specific metabolite) by an engineered acetogen. Rather, the novel aspects are: 1) reducing ANP mixotrophy to practice and demonstrating in both standard and enhanced modes of operation, and 2) showing that ANP mixotrophy is capable of being deployed in a broad range of acetogenic hosts with minimal to no carbon catabolite repression.

As mentioned, the first novel aspect is reducing to practice and demonstrating the value of acetogenic mixotrophy to greatly enhance sugar mass yields to products with and without the addition of exogenous reductant. Acetone production is but one example that demonstrates the breakthrough advantages and utility of this technology when applied to industrial biotechnology. We apologize for the major oversight of not expanding upon the high cell density continuous fermentation system in our originally submitted version of the manuscript. This was because the previous manuscript version was a very short format and transferred without editing to Nature Communications. So details and discussion were severely limited. We have now addressed this issue in the revised manuscript, which demonstrates acetone yields, productivities, and titers well beyond the current state of the art.

Secondly, we dedicated copious research and effort towards understanding catabolite repression, interrogating mixotrophy versus enhanced mixotrophy, and evaluating the potential for mixotrophy in four acetogenic strains that are well cited in the current literature and phenotypically diverse from one and other.

The authors proved their hypothesis that syngas enhanced mixotrophy would deliver higher incorporation of syngas into products while avoiding the formation of undesired products like acetate.

Nevertheless the authors fail to mention previous work using acetogen *C. aceticum* where the native pathway from *C. acetobutylicum* was cloned to produce acetone independently from acetate.

The reviewer did not provide a specific reference. We assume the reviewer refers to the following review: Schiel-Bengelsdorf, B. & Durre, P. Pathway engineering and synthetic biology using acetogens. *FEBS letters* **586**, 2191-2198, doi:10.1016/j.febslet.2012.04.043 (2012). We now cite this reference, reluctantly so because as in any review manuscript, there are no primary data shown, and although a strategy is described, there are not detailed methods. The following PhD thesis is cited instead:

- Lederle, S. *Heterofermentative Acetonproduktion* Ph.D. thesis, University of Ulm, Germany, (2011).

Please note that we have been aware of this work, but have serious reservations citing it in our manuscript for reasons described below. The FEBS Letters review mentions the formation of 8-9 mg/L acetone, not a significant titer. The PhD thesis is in German. To our knowledge there is no English translation available, and this is NOT a peer-reviewed reference that is widely available to the scientific community. Finally, this thesis and this research appears to have been supported by Evonik, and subsequently is the subject of an Evonik patent. We again have significant reservations in including citations from patents, as they are not references scrutinized by peer review.

Again, in spite of our reservations, we now include the FEBS Letter reference at the request of Reviewer #2, but we AGAIN caution that the focus of this paper is now being skewed towards an applied tone only for the production of acetone, which is NOT our intended focus.

The authors also fail to mention a relevant paper review about pathway engineering and synthetic biology tools using acetogens, which covers a whole section for the production of acetone by acetogens (B. Schiel-Bengelsdorf and P. Durre, 2012, Pathway engineering and synthetic biology using acetogens, *Synthetic Biology*, 586, 2191-2198). In that review, previous works engineering acetogens to produce acetone up to 8 mg/L are described which were not mentioned in this manuscript, thus, proof of principle for using recombinant acetogens growing on CO₂/H₂ gas mixtures and producing important bulk chemicals has already been provided, so the novelty is using CLJ as a proof of principle instead of *C. aceticum*.

As mentioned, we now cite: Schiel-Bengelsdorf, B. & Durre, P. Pathway engineering and synthetic biology using acetogens. *FEBS letters* **586**, 2191-2198, doi:10.1016/j.febslet.2012.04.043 (2012).

Again, but strongly disagree with Reviewer #2's insistence that the novelty and point of our paper is regarding the recombinant production of metabolites in acetogens. Moreover, we are comparing mixotrophy to autotrophy and all references suggested by Reviewer #2 focus solely on autotrophy. Therefore, we claim significant novelty.

Nevertheless, the authors do not give any final acetone yield in mg/L and very few quantitative comparisons are given with previous works in literature, which really undermines the manuscript. Also, more rigorous results are needed to completely backup some claims from the authors.

We do regret not extending the content of our manuscript, as this was a transfer for another Nature journal where we had submitted a short form manuscript. Accordingly, we have now revised to contain significantly more information and data on the high cell density continuous fermentation work. We now include titers (g/L), yields (wt%), productivities (g/l/hr), cell densities (OD600nm) and operating parameters for the fermentation approach.

Also, some results were only repeated twice, which I believe is not enough.

We believe Reviewer #2 is referring to the biological replicate experiments shown in Figure 1. The fact that we were able to accomplish such reproducible results, as evidenced by the tight standard deviations, provides sufficient support for the conclusions in our opinion. From a practical perspective, specialty made, fully ¹³C-labeled gases are INCREDIBLY expensive. 300 liters of one atmosphere labeled gas of this kind costs nearly \$15,000. Accordingly, conducting these experiments are not a trivial and inexpensive task.

Based on this, I would not recommend the manuscript for publication in Nature Communications.

My other specific comments to improve the paper would be as follows:

- 1) I would add a nomenclature section to enhance the reading experience. To our understanding, a nomenclature section is not common practice for Nature Communication articles. All acronyms are fully spelled out before using, aside from rare occasions when using standard biology nomenclature such as NAD(P)H. Accordingly, we intend to keep our manuscript consistent with Nature Communication articles and not include a nomenclature table.
- 2) Adding a sentence in the introduction describing the main characteristic of acetogens would be useful

Now that we have more text to use for this manuscript, we are happy to accommodate this request.

3) Adding a sentence in the introduction mentioning why acetone is an industrially relevant product to synthesize using microorganisms would be appropriate.

We agree that this is a good suggestion and added this on page 5, lines 82-85.

4) In the introduction, it would be worth to mention if any acetogens naturally produce acetone? Or if they have been engineered to produce acetone?

We have included this information in the subsection focused on acetone production (pages 7, lines 121-128). However, we do not wish to expand so much upon acetone in the introduction, as this is not the focus or the basis for the novelty of our manuscript.

5) I was a little confused in this: In line 71 it is mentioned that *C. Ljungdahlii* was selected to determine if it didn't suffer from catabolite repression of the Wood-Ljungdahl pathway (WLP) in the presence of the preferred sugar fructose (in comparison with *C. aceticum* and *Blautia cocoides* which do suffer catabolite repression). The authors seem to conclude CLJ doesn't suffer from catabolite repression, based on the comparison of either transcript and protein levels with an auxotroph control. Nevertheless the results of the auxotroph control are not shown in Fig S1a (transcription studies), it would be good to add them.

We conclude there is no catabolite repression based on the transcriptional, proteomic, and metabolic data. The ^{13}C metabolite labeling analysis is the major experimental evidence for the lack of catabolite repression, while the transcriptional and proteomic data support this evidence. The fold changes presented in Fig. S1a are ratios of either mixotrophy/autotrophy or syngas-enhanced mixotrophy/autotrophy. These fold changes are not absolute values but rather relative values to the control case (i.e., autotroph). We have revised the Methods section and the figure legend to make this and all other experimental details more clear.

6) Also, relevant to comment 4, in Figure S1b, only one protein of the whole WLP was studied using western blot. Why was this enzyme selected and why were not the other enzymes studied too? How do the authors know there was no translational repression of the other 3 enzymes ?

The antibody was against the CODH subunit of the ACS/CODH unit, which is the final enzyme in the WLP which brings together the two branches. This enzyme is essential for the proper function of the WLP. While we would have liked to test all of the enzymes in the WLP, these are custom antibodies which are both expensive and time-intensive to make. Our conclusion of the lack of catabolite repression is based on all the data, transcriptional, proteomic, and ^{13}C metabolite analysis, and based on the positive results from all these assays, we conclude there is no catabolite repression.

7) Relevant to comment 4 and 5, if it was indeed proven that CLJ did not suffer from catabolite repression, adding one sentence would be adequate to compare with results of the repressed strains? (maybe add an hypothesis of why CLJ may be different ?)

This section has been slightly revised to highlight past research where two acetogens

have shown catabolite repression and two acetogens have not shown catabolite repression. Based on the data from our studies, we cannot make a firm hypothesis on why some acetogens show catabolite repression and some do not, but one explanation could be different environmental niches and evolutionary histories.

8) In line 71, what was the concentration of fructose used in the labeling experiments?. In the materials and methods it says 5 g/l (lines 269 and 274), but in figure 1 c and d, it seems fructose was 10 g/l at time 0.

The reviewer is correct that the starting concentration is 10 g/l and the expanded Methods section has been revised to make this more clear (page 15, lines 304-305).

9) In Line 73 it is mentioned that the syngas composition is (CO:CO₂: H₂: N₂: 55:10:20:15) : Why was this specific component concentration of syngas selected? Was it based on some previous optimization?

The syngas is the same composition as a mixture used previously with CLJ (Phillips, J.R., et. al. 1994. *Appl Biochem Biotechnol* 45/46:145-57), except the Ar was replaced with N₂. It was not based on any previous optimization.

10) In Line 75 it is mentioned: "CLJ incorporated a surprisingly large percentage of labeled gas into products". Any hypothesis why this marked preference for syngas instead of fructose? How does it compare in literature with other mixotrophic acetogens?

It is not clear why there is a preference for syngas over fructose, and this is part of the future work to be carried out. This is also getting more to the novelty of the paper. There are no focused, detailed and specifically thought-out studies to interrogate gas and sugar consumption with unequivocal results based upon ¹³C labeling.

11) In line 84 and Figure 1 c and d, Why when using *C. autoethanogenum* (CAU) all fructose is consumed but not with CLJ? Probably this is related to the comment 9. It would be nice to further explain, as this seems to be an industrially relevant feature from CLJ.

This has been expanded on in the discussion (see pages 13 and lines 259-272). As we say, it is not clear why this occurs and may be related to an inhibitory concentration of acetate, though this needs further study.

12) In line 84 and figure 1c and d: How was growth yield different between the two microorganisms in terms of growth yield etc?

If the reviewer is referring to the OD_{600nm} of the two strains, CAU reached an OD_{600nm} of ~3.0 and CLJ reached an OD_{600nm} of ~1.5. The difference is most likely related to the higher amount of fructose consumed by CAU versus CLJ.

13) In line 75, How does the labeled gas incorporation into acetate compares with other labeling works made for acetogens mixotrophes or mixotrophes in general?

There is only one other ¹³C-labeling study we are aware of (Yun, S., et. al. 2013. *Bioprocess Biosyst Eng* 36:591-95). This study also used CLJ and grew it on 5 g/l fructose with a ¹³CO headspace (either 0%, 60% CO/40% H₂, or 90% CO/10% H₂). However, this study only quantified the ¹³C incorporation into biomass and did not look

specifically at any metabolites, therefore a direct comparison cannot be made. In addition, the headspace gas compositions were significantly different, further complicating the comparison. The headspace gas used in the other ^{13}C study is highly reduced (only CO and H_2), and it's not clear the effect this has on gas uptake, which is another direction of future work. The bottom line is that there are no other detailed ^{13}C -based metabolic studies and this is disappointing for the field. Our data are not only unique in this sense, but they are unlikely to be followed up soon by others given the high cost and effort necessary to do so.

14) In line 75 and line 81 and Figure Did the labeling for ethanol was also measured? How does it compare with acetate?

The labeling of ethanol was not directly measured, but should be similar to acetate since they both come from acetyl-CoA. This has been made clear in the text (page 6, lines 109-111).

15) In line 84 and figure 1c and d : any hypothesis would CAU produce more ethanol since both CAU and CLJ have the enzyme AAD to produce ethanol from Acetyl -CoA ? It has been known in the literature that CAU in general produces more ethanol than CLJ, despite their high degrees of similarity. Since this study was not focused on this, we cannot offer a hypothesis on this.

16) In line 93, it was found that after knocking out the secondary alcohol dehydrogenase (SADH) coding gene, the resulting strain produced acetone and acetate but also high quantities of 3-hydroxybutyrate. Why was the gene of the enzyme 2,3-butanediol dehydrogenase not knocked out too to increase acetone yield?

We are completing that work but do not intend to report those results in this manuscript. We can readily quantify 3-HB, so we are able to close mass balances, calculate mass yields, etc. Moreover, acetone production is NOT the focus of this paper.

17) Line 95: Was it possible to measure the total production in g/L of acetate, and 3-HB? Only carbon fractions are presented in Fig B and C.

These titers have now been added to the main text, along with a supplemental table for all metabolites (Supplemental Table S1).

18) In line 96: " 3-HB is presumably produced from the native 2,3- butanediol dehydrogenase (2,3BDH) acting upon the intermediate acetoacetate", adding a reference would be adequate.

We cannot find a reference in the literature for this, however we think it is appropriate to hypothesize how 3-hydroxybutyrate is formed. Since no 3-HB is produced by the host, 3-HB must derive from acetone or an intermediate of the synthetic pathway, and acetoacetate is the only one where the chemistry makes sense.

19) In Line 99, while using the fermentor, what was the final cell density and final yield in g/L of acetone and 3-HB?

We have added considerably more details, description and discussion of the continuous fermentation work. All data is provided either in Fig. 3 or supplemental Figs. S2 or S3.

20) Line 105. The results of 20% and 40% H₂ look statistically the same, why didn't the authors try using 10%, which would have favored the production of acetone?

This is a reasonable hypothesis, but our thermodynamic modeling does not support the conjecture that less hydrogen favors acetone. In fact, minimizing the concentration of CO₂ actually appears to have the greatest thermodynamic benefit to enhancing flux to acetone. CO₂ is a by-product of acetone production, thus the mass action ratio of the chemical equilibrium better supports the hypothesis that reducing CO₂ activity is preferred. Accordingly, in batch serum bottles, we believe CO₂ concentration has the greater impact on product profile. These are topics for future evaluation and study but beyond the scope of this paper.

21) In line 124, "the syngas mixture affected the product profiles with all strains producing a greater portion of reduced products (Supplemental Fig. S2)". It would be interesting to add one more sentence to explain why? It is especially interesting to see in Figure S2 the shift from microorganism ELM to produce butyrate instead of ethanol. A sentence has been added explaining why the shift in metabolites occurred (page 10-11, lines 211-215). As far as we know, the ELM strain we used, *E. limosum* DSM-20543, is not known to produce ethanol. Even under highly reduced conditions (e.g., methanol as a substrate), butyrate is the preferred reduced product with no ethanol production.

22) In line 140, it would be interesting to compare the maximum biological mass yields predicted in Figure 3b under different mixotrophic condition with results from literature? This only applies to acetate production, which does reach nearly the theoretical maximum. This comment again points to the novelty of our work, which is that ANP mixotrophy is not a studied field and not exploited for its industrial biotechnology advantages. Accordingly, we would like to keep the table as is because attempting to populate with literature data would create a very sparse table.

23) In line 288, where de genes of thiolase, acetate/acetoacetate CoA-transferase and acetoacetate decarboxylase codon optimized?

No, the genes were not codon optimized. The source organisms were also clostridial, and no optimization was needed.

24) In line 290, how was the transformation achieved?

The Methods have now been revised to include this.

25) The pathway from Figure 1a is rather over simplified, something like the pathway shown in Figure 1a of the paper from Kopke et al. 2010 (the author is cited in this manuscript) would make it easier for the reader to understand the paper. Specially when enzymes like ACS/CODH, ACS alpha, ACS Beta, CFeSP alpha, CFeSP beta are mentioned in the paper but the full name is not given

The pathway in Fig. 1a was kept simple purposefully to avoid overcomplicating the pathway and to help broaden the audience for the manuscript. More detailed figures are presented in the reviews of mixotrophy which we cite (Tracy, et. al. 2012. and Fast, et al. 2015), and we did not feel this manuscript would benefit from reproductions of them.

However, the reviewer's point regarding the enzyme names is well taken, and Fig. S1 has been revised to include a small schematic of the WLP to aid the reader in understanding the function of these enzymes.

26) In the discussion or conclusions, it would be relevant to add a sentence mentioning the key challenges to be addressed in order to improve yields using mixotrophy?
This has been addressed in the revised discussion section.

Reviewer #3 (Remarks to the Author):

Jones et al. show that anaerobic, non-photosynthetic mixotrophy can both increase product conversion yields and reduce overall CO₂ emissions from fermentation processes. As a proof-of-concept, they engineered *Clostridium ljungdahlii* to produce acetone at a high mass yield and demonstrated that when enough reductant (e.g., H₂) is provided no CO₂ was produced. They also show that mixotrophy is a general trait among most acetogens, and present an experimentally-validated model to predict yield increases for a variety of bioproducts of industrial interest.

Although it is a very nice piece of work with interesting concepts, the main concern of the manuscript is that most of the experiments regarding acetone production by *C.*

ljungdahlii cannot, in their current state of description, be reproduced:

The very brief description of experimental details was because the submitted manuscript version was a very short format and transferred without editing to Nature Communications. So details and discussion were severely limited. We have now expanded upon the methods and results, particularly for the high cell density, continuous fermentation.

1) How was the deletion of *sadh* made by homologous recombination (replicative/non-replicative plasmid, use of PCR product? How was the DNA introduced in *C. ljungdahlii*?)

We have now added all the necessary details in the Methods section. Briefly, the deletion was accomplished by double crossover homologous recombination (also referred to as allelic exchange), with a replicating plasmid that used dual antibiotic resistance to select for double crossover events via replica plating. The DNA was introduced into CLJ via standard electroporation protocols, now referenced in the text.

2) How was the pTCTA plasmid constructed? Was the *thlA* promoter from *C. acetobutylicum* conserved in the pSOS95 or was it replaced by the *thlA3* promoter from *C. kluyveri*?

We have now added more detail in the Methods section, specifically on page 15-16.

3) How was the high cell density culture performed? What were the bleeding and dilution rates? What was the mass yield of acetone formation (currently only the mass yield of acetone+3HB is given)? Although the mass yield is an important parameter, titer and productivity are also key for an industrial process and should also be provided.

We have now added more detail in the Methods section, specifically on pages 16-17.

Other comments:

-Line 90 It is claimed that *sadh* (deleted in the acetone producing strain) is naturally responsible for 2, 3 butanediol production. Then line 94 it is said that "3-HB is presumably produced from the native 2,3-butanediol dehydrogenase acting upon the intermediate acetoacetate ». How can SadH reduced acetoacetate if the encoding gene has been deleted?

We apologize this was not more clear. We have now expanded this section to better explain this (see page 7, lines 136-140). To make 2,3-butanediol, CLJ uses both a SADH

and a 2,3BDH. We deleted the SADH but did not delete 2,3BDH. We believe it is the activity of 2,3BDH that is acting upon acetoacetate to make 3-HB.

-Mixotrophic growth with high mass yield of product formation has already been described for *Eubacterium limosum* and the reference should be added (Loubière, P., GRos, E. V. E. L. Y. N. E., Paquet, V., & Lindley, N. D. (1992). Kinetics and physiological implications of the growth behaviour of *Eubacterium limosum* on glucose/methanol mixtures. *Microbiology*, 138, 979-985.)

This was an oversight on our part, and this reference has been now added.

Reviewer #1 (Remarks to the Author)

Authors did nice job addressing the comments. I have no further comments.

Reviewer #3 (Remarks to the Author)

The authors have made every attempt to address the open issues raised by me and the other reviewers. I have no further remarks and am fully satisfied with all comments by the authors. I recommend this fine paper for publication in Nature Communication as it stands.